# Evaluation of hematological parameters alterations in different waves of COVID-19 pandemic: A cross-sectional study

Javad Charostad[1,2], Mohammad Rezaei Zadeh Rukerd[3], Azadeh Shahrokhi[4], Faezeh Afkhami Aghda[2], Yaser ghelmani[5,6], Pouria Pourzand[7], Sara Pourshaikhali[8], Shahriar Dabiri[9], Azam dehghani[10], Akram Astani[1,2], Mohsen Nakhaie[3]*, Ehsan Kakavand[11]

1 Department of Microbiology, Faculty of Medicine, Shahid-Sadoughi University of Medical Sciences, Yazd, Iran, 2 Student Research Committee, Shahid Sadoughi University of Medical Sciences, Yazd, Iran, 3 Gastroenterology and Hepatology Research Center, Institute of Basic and Clinical Physiology Sciences, Kerman University of Medical Sciences, Kerman, Iran, 4 Department of Physiology and Pharmacology, Afzalipour School of Medicine, Kerman University of Medical Sciences, Kerman, Iran, 5 Department of Internal Medicine, Shahid Sadoughi University of Medical Sciences, Yazd, Iran, 6 Clinical Research Development Center of Shahid Sadoughi Hospital, Shahid Sadoughi University of Medical Sciences, Yazd, Iran, 7 Department of Emergency Medicine, Kerman University of Medical Sciences, Kerman, Iran, 8 Pathology and Stem Cell Research Center, Kerman University of Medical Sciences, Kerman, Iran, 9 Department of Pathology, Afzalipour Faculty of Medicine, Kerman University of Medical Sciences, Kerman, Iran, 10 Department of Medical Virology, School of Medicine, Ahvaz Jundishapur University of Medical Sciences, Ahvaz, Iran, 11 Department of Virology, School of Public Health, Tehran University of Medical Sciences, Tehran, Iran

* mohsennakhaee1367@gmail.com

**Data Availability Statement:** All relevant data are within the paper and its Supporting Information files.

## Abstract

### Background

The occurrence of variations in routine hematological parameters is closely associated with disease progression, the development of severe illness, and the mortality rate among COVID-19 patients. This study aimed to investigate hematological parameters in COVID-19 hospitalized patients from the 1st to the 5th waves of the current pandemic.

### Methods

This cross-sectional study included a total of 1501 hospitalized patients with laboratory-confirmed COVID-19 based on WHO criteria, who were admitted to Shahid Sadoughi Hospital (SSH) in Yazd, Iran, from February 2020 to September 2021. Throughout, we encountered five COVID-19 surge waves. In each wave, we randomly selected approximately 300 patients and categorized them based on infection severity during their hospitalization, including partial recovery, full recovery, and death. Finally, hematological parameters were compared based on age, gender, pandemic waves, and outcomes using the Mann-Whitney U and Kruskal-Wallis tests.

**Funding:** The authors received no specific funding for this work.

**Competing interests:** The authors declare that they have no competing interests.

**Abbreviations:** SARS-CoV-2, Severe acute respiratory syndrome coronavirus 2; CBC, Complete blood count; WBC, White blood cell; RBC, Red blood cell; PLT, Platelet; NLR, Neutrophil-to-lymphocyte ratio; RT-PCR, Reverse transcription polymerase chain reaction; WHO, World health organization; SSH, Shahid sadoughi hospital; UNESCO, United nations educational, scientific and cultural organization; SSU, Shahid sadoughi university; Hct, Hematocrit; Hb, Hemoglobin; MCH, Mean corpuscular hemoglobin; MCHC, Mean corpuscular hemoglobin concentration; MCV, Mean corpuscular volume; PDW, Platelet distribution width; RDW, Red blood cell distribution width; ESR, Erythrocyte sedimentation rate; HIS, Health information system; NET, Neutrophil extracellular trap; SOD1, Superoxide dismutase 1; G6PD, Glucose-6-phosphate dehydrogenase; Prxs, Peroxiredoxin.

## Results

The mean age of patients (n = 1501) was 61.1±21.88, with 816 (54.3%) of them being men. The highest mortality in this study was related to the third wave of COVID-19 with 21.3%. There was a significant difference in all of the hematological parameters, except PDW, PLT, and RDW-CV, among pandemic waves of COVID-19 in our population. The highest rise in the levels of MCV and RDW-CV occurred in the 1st wave, in the 2nd wave for lymphocyte count, MCHC, PLT count, and RDW-SD, in the 3rd wave for WBC, RBC, neutrophil count, MCH, and PDW, and in the 4th wave for Hb, Hct, and ESR ($p < 0.01$). The median level of Hct, Hb, RBC, and ESR parameters were significantly higher, while the mean level of lymphocyte and were lower in men than in women ($p < 0.001$). Also, the mean neutrophil in deceased patients significantly was higher than in those with full recovered or partial recovery ($p < 0.001$).

## Conclusion

The findings of our study unveiled notable variations in hematological parameters across different pandemic waves, gender, and clinical outcomes. These findings indicate that the behavior of different strains of the COVID-19 may differ across various stages of the pandemic.

## 1. Introduction

The COVID-19, an infectious disease caused by the severe acute respiratory syndrome coronavirus 2 (SARS-CoV-2), initially emerged in Wuhan, China, in late December 2019, rapidly evolving into a significant global health crisis [1, 2]. The impact of COVID-19 on a global scale has been significant, resulting in over 766 million reported infections and more than 6.9 million deaths worldwide [3, 4]. Iran has experienced a notable impact from COVID-19, with more than 7.6 million confirmed cases of infection and nearly 146,000 recorded deaths [5]. COVID-19 primarily affects the respiratory system, but it can also target other vital systems, including the cardiovascular, gastrointestinal, nervous, immune, and hematopoietic systems [6, 7]. This multi-system involvement contributes to increased morbidity and mortality rates throughout the course of the disease [8]. Notably, older adults and individuals with underlying conditions such as cardiovascular disease, diabetes mellitus, chronic respiratory diseases, and cancer are particularly susceptible to a higher risk of fatal outcomes [8]. The identification of potential risk factors that can predict the progression of the disease holds significant value in terms of its practical utility [8, 9]. Consequently, the development of accurate risk assessment tools is crucial in light of the importance of identifying the condition of COVID-19 patients, considering the wide range of disease progression from mild cases with a favorable prognosis to severe cases with an unfavorable prognosis [8, 10, 11].

Numerous risk assessment tools have been implemented to predict the prognosis of COVID-19 [8, 12]. However, it is believed that the current risk assessment models have a substantial risk of bias due to inherent methodological limitations [13]. These models are often time-consuming and costly to implement, which limits their widespread application, particularly in regions with limited medical resources [12]. On the other hand, alternative approaches have been proposed, advocating the use of clinical and laboratory parameters as simple, cost-

effective risk prediction tools for COVID-19, aiming to assist frontline physicians in optimizing medical interventions [14, 15].

Hematological parameters have been extensively evaluated in predicting the severity of COVID-19 patients [16, 17]. The complete blood count (CBC) is a routinely used as the most prevalent laboratory test employed universally across different regions to investigate the correlation between hematological parameter levels and the clinical status and prognosis of COVID-19 patients [18–20]. Researchers have identified specific laboratory findings that hold prognostic potential in COVID-19 patients. These findings include alterations in white blood cell (WBC), red blood cell (RBC), platelet (PLT), lymphocyte, and neutrophil counts, as well as the neutrophil-to-lymphocyte ratio (NLR) and morphological changes in RBCs [19, 21].

The primary objective of the present study is to provide fundamental insights into COVID-19 and its potential variations across different waves. Specifically, the study aims to compare the hematological parameters of COVID-19 patients between various waves, from the initial detection of COVID-19 in Iran up to the fifth wave. The results obtained from this research hold particular significance in optimizing clinical decision-making, particularly in countries facing substantial shortages of medical resources, particularly in reducing mortality rates.

## 2. Material and methods

### 2.1. Study design and participants

In this cross-sectional study, a total of 1501 hospitalized patients diagnosed with COVID-19 were enrolled. The confirmation of COVID-19 was based on reverse transcription polymerase chain reaction (RT-PCR) testing, following the criteria set forth by the World Health Organization (WHO). Participants with missing data in their hematological parameters and those with negative RT-PCR test results were excluded from the study. The flowchart outlining the participant selection process for this study is illustrated in Fig 1.

The patients were admitted to Shahid Sadoughi Hospital (SSH) in Yazd, Iran, from February 2020 to September 2021, encompassing all five waves of COVID-19 surge. Yazd, located in the Central geographic region of Iran, serves as the capital of Yazd province. This city holds the distinction of being recognized as a world heritage site by the united nations educational, scientific and cultural organization (UNESCO). Renowned for its hot desert climate, Yazd is considered the driest major city in Iran. The annual precipitation in Yazd is less than 50 millimeters, and during summer, temperatures frequently soar above 40˚C (104˚F) under intense sunlight with low humidity (The map of Yazd city is provided as a S1 File) [22, 23]. Shahid Sadoughi University (SSU) is the sole public medical university in this province, with SSH serving as its primary affiliated hospital. The study protocol was in line with human subject protection regulations, approved by the Research Ethics Committee of Shahid-Sadoughi University of Medical Sciences (IR.SSU.REC.1400.207), and written informed consent was obtained from each participant.

The present study investigates the impact of five distinct waves of the COVID-19 pandemic on the study population, as visualized in Fig 2. With the exception of the first wave, which included 301 patients, each wave entailed the random selection of 300 COVID-19 patients. Subsequently, these patients were categorized into three groups based on the severity of the disease during their hospitalization: partial recovery, full recovery, and death. Patient-related data, including essential information such as age, gender, pandemic wave, clinical outcome, and various hematological parameters, including WBC, RBC, PLT, lymphocyte, and neutrophil counts, as well as hematocrit (Hct), hemoglobin (Hb), mean corpuscular hemoglobin (MCH), mean corpuscular hemoglobin concentration (MCHC), mean corpuscular volume (MCV), platelet distribution width (PDW), red blood cell distribution width (RDW)-CV,

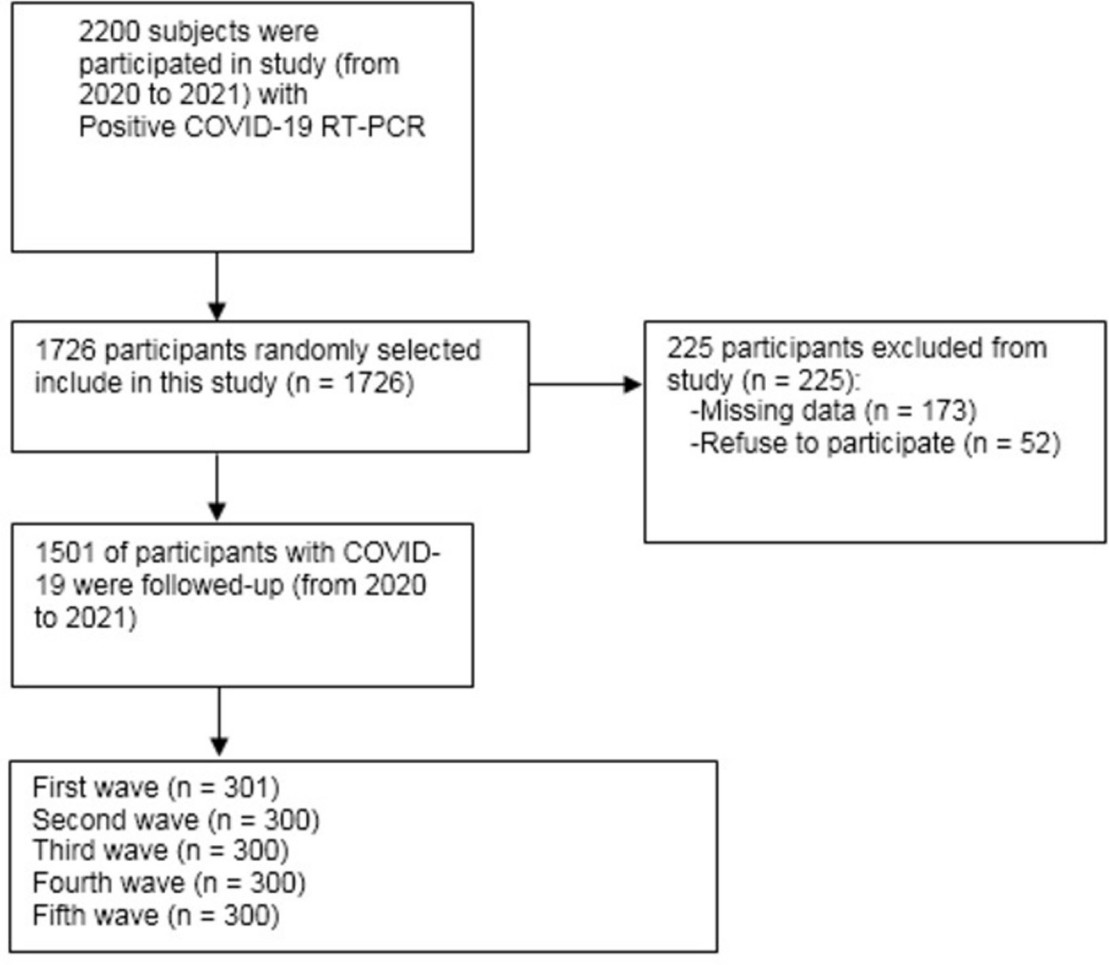

**Fig 1. The flowchart of the participants in the study.**

RDW-SD, and erythrocyte sedimentation rate (ESR), were extracted from Health Information Systems (HISs).

## 2.2. Statistical analysis

Quantitative patient characteristics were reported as mean (standard deviation, SD), while qualitative variables were presented as frequency (percentages). The numerical variables exhibited a normal distribution. To compare the variations in hematological parameters based on age, gender, pandemic waves, and outcomes, we utilized the Mann–Whitney U test and Kruskal-Wallis test. The data analysis was conducted using SPSS version 26, with a significance level (alpha) set at 0.05. A P value below 0.05 was considered statistically significant for all conducted tests.

## 3. Results

Among the total number of patients involved in this study (N = 1501), 816 (54.3%) were male, while 685 (45.7%) were female. The demographic characteristics and clinical outcomes of

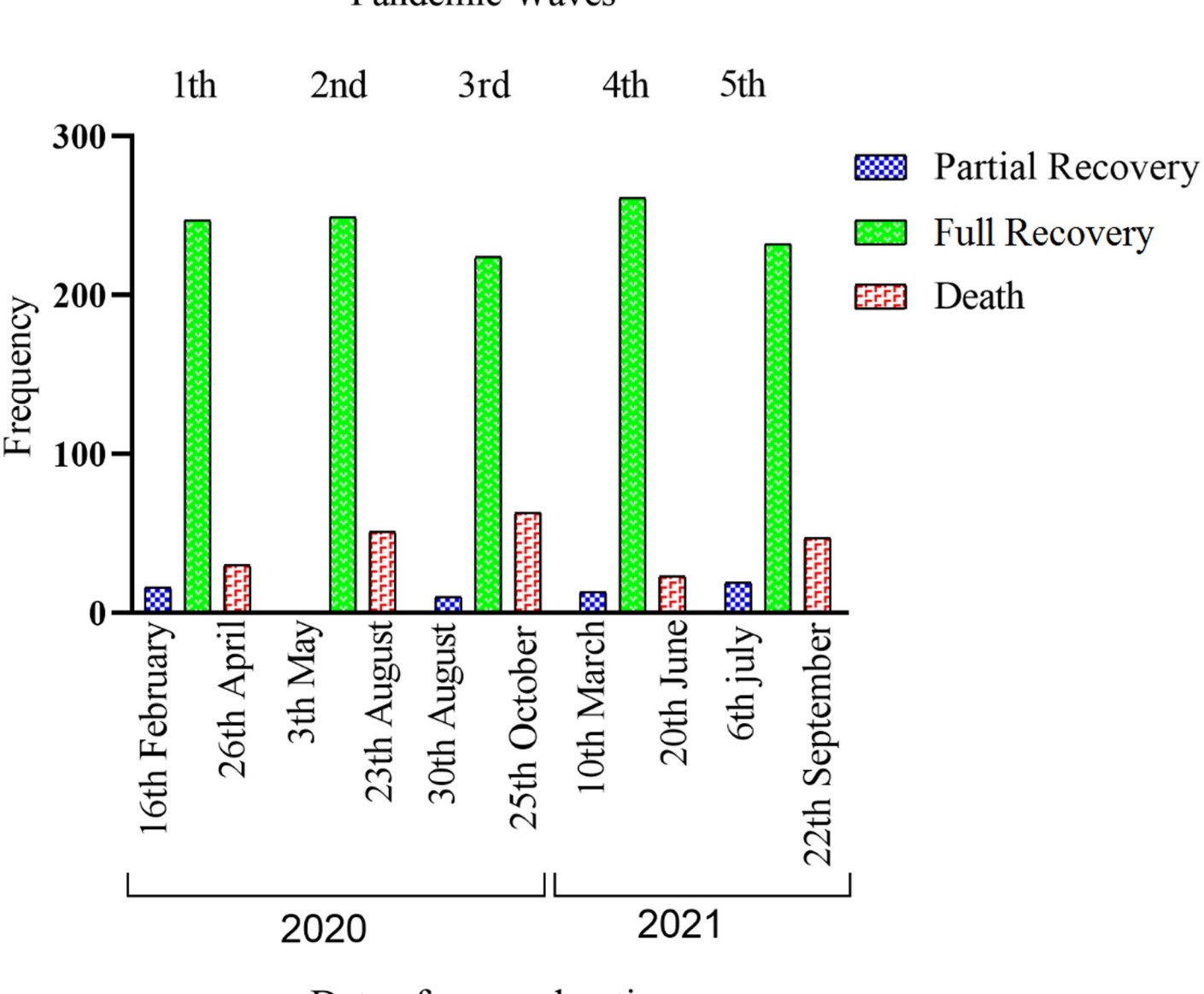

**Fig 2. The time frame of each of the COVID-19 waves in Iran and its relationship with the outcomes.**

hospitalized patients with COVID-19 from the beginning of the COVID-19 pandemic until the fifth wave are presented in Table 1.

The mean age of the patients was recorded as 61.1±21.88 years. The median age of hospitalized patients exhibited a significant difference among the waves, with the 4th wave having a lower median age of 59 years, while the 3rd wave had a higher median age of 72 years compared to the other waves. The frequency distribution of hospitalized cases of COVID-19 revealed that during the 3rd wave, there was a higher percentage of infection among older adult patients aged 65 years and above (59.7%), whereas in the 4th wave, a higher proportion of younger patients were infected (54.2%).

**Table 1. Demographic characteristics and in-hospital outcome in patient with COVID-19 during five pandemic waves.**

|  | 1th | 2nd | 3rd | 4th | 5th | P-values |
|---|---|---|---|---|---|---|
| **Age** |  |  |  |  |  |  |
| Mean ± SD | 59.8±24.86 | 65.83±17.13 | 62.14±24.19 | 56.24±21.57 | 61.5±19.75 |  |
| Median (IQR) | 70 (35.25–82) | 67.5 (48–80) | 72 (38–81) | 59 (37–74) | 65 (46.75–77) | *<**0.001** |
| **Age: Distributions** |  |  |  |  |  |  |
| <65 | 137 (45.5%) | 150 (50%) | 121 (40.3%) | 163 (54.3%) | 146 (48.7%) |  |
| ≥ 65 | 164 (54.5%) | 150 (50%) | 179 (59.7%) | 137 (45.7%) | 154 (51.3%) | **0.012**\* |
| **Gender** |  |  |  |  |  |  |
| Male | 161 (53.6%) | 160 (53.3%) | 171 (57%) | 159 (53%) | 165 (55%) |  |
| Female | 140 (46.4%) | 140 (46.7%) | 129 (43%) | 141 (47%) | 135 (45%) | 0.87 |
| **Outcomes** |  |  |  |  |  |  |
| Full Recovery | 251 (83.3%) | 249 (83.1%) | 225 (75%) | 262 (87.3%) | 233 (77.6%) |  |
| Partial Recovery | 17 (5.6%) | 0 (0) | 11 (3.7%) | 14 (4.7%) | 20 (6.7%) |  |
| Death | 33 (10.9%) | 51 (16.9%) | 64 (21.3%) | 24 (8%) | 47 (15.7%) | *<**0.001** |

*P<0.01, SD: Standard Deviation, IQR: Interquartile range

In all waves of COVID-19, a higher number of men were hospitalized compared to women; however, no significant relationship was observed between gender and hospitalization rates. The mortality rate showed a significant increase from the 1st to the 3rd wave (10.2% vs. 21.2%), but there was a decline from the 4th to the 5th wave (7.7% vs. 3.2%). Notably, the 3rd wave of COVID-19 within our population recorded the highest mortality rate of 21.2%, as depicted in Fig 2.

The results of Kruskal Wallis test indicated a statistically significant difference among the pandemic waves of COVID-19 in our population for all hematological parameters, except for PDW, PLT count, and RDW-CV, as presented in Fig 3. The highest rise in the levels of MCV and RDW-CV occurred in the 1st wave, in the 2nd wave for lymphocyte count, MCHC, PLT count, and RDW-SD, in the 3rd wave for WBC, RBC, neutrophil count, MCH, and PDW, and in the 4th wave for Hb, Hct, and ESR ($p < 0.01$). The findings suggest that the different waves of the pandemic had diverse effects on the hematological parameters. For instance, while RBC count, level of Hct, and Hb remained within the normal range across all waves, the most significant increase in these parameters was observed during the 3rd and 4th waves, while the lowest increase occurred during the 1st wave. On the other hand, the 3rd wave had the greatest impact on leukocytosis (increase in white blood cell count), neutrophilia (increase in neutrophil count), and lymphopenia (decrease in lymphocyte count), whereas the 1st wave had the least impact on these parameters.

A significant association was observed between hematological parameters and age groups. Specifically, the median values of Hct, Hb, and RBC, lymphocyte, PLT counts, were significantly higher in younger individuals compared to older individuals. Conversely, the mean values of WBC counts, neutrophil counts, and ESR were significantly higher in the older adult population compared to the younger population. These associations were found to be statistically significant, with a P value of less than 0.01, as illustrated in Fig 4.

The results of the Mann-Whitney U test demonstrated significant differences between men and women in terms of RBC count, Hct, Hb, and ESR. Specifically, the median levels of these parameters were found to be significantly higher in men compared to women ($p < 0.001$), while the median level of lymphocyte and PLT count were lower in men than women ($p < 0.01$) (Table 2).

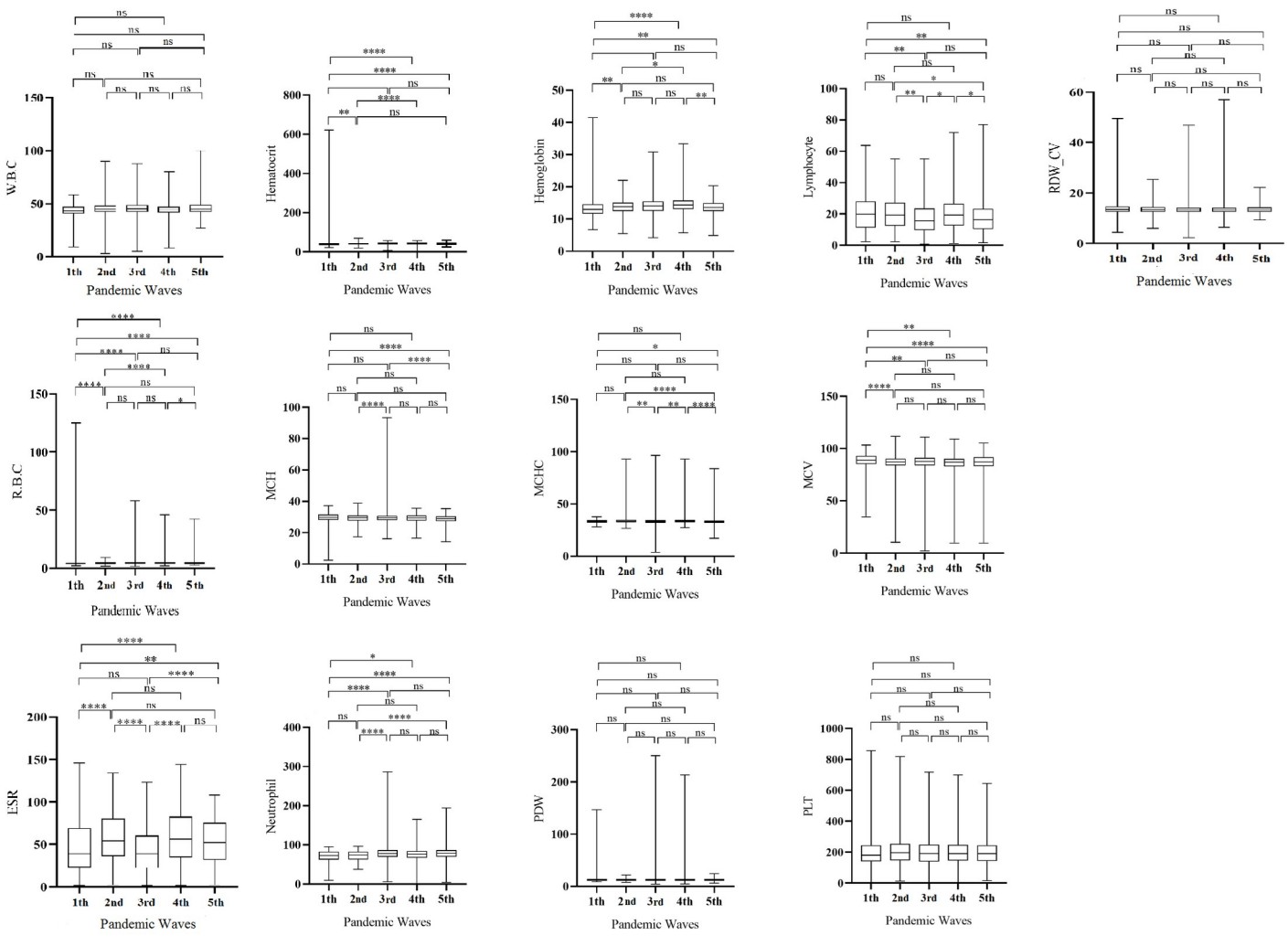

**Fig 3. The results of Kruskal-Wallis test in all of the hematological parameters.**

In the 3rd wave, most of the hematological parameters, such as leukocytosis, neutrophilia, lymphopenia, increased RBC count, PLT count, Hb level, and RDW-SD, underwent more changes compared to other waves, indicating a worse outcome in patients. The results of the Kruskal-Wallis test indicated that the mean of WBC, RBC, lymphocyte, and PLT counts, as well as Hb level, was significantly lower in deceased patients compared to fully recovered or partially recovered patients ($p < 0.05$). Furthermore, the mean value of neutrophil counts was found to be significantly higher in deceased patients compared to those who had either partial or full recovery ($p < 0.001$). There were no significant differences in the median of Hct or ESR in deceased patients compared to fully recovered or partial recovered patients ($p > 0.05$) (Table 3).

## 4. Discussion

In this study, we conducted an evaluation of readily accessible and routine hematological parameters in patients diagnosed with COVID-19 who sought medical care at Shahid Sadoughi Hospital (SSH) in Yazd, Iran. The study period extended from November 2019 to

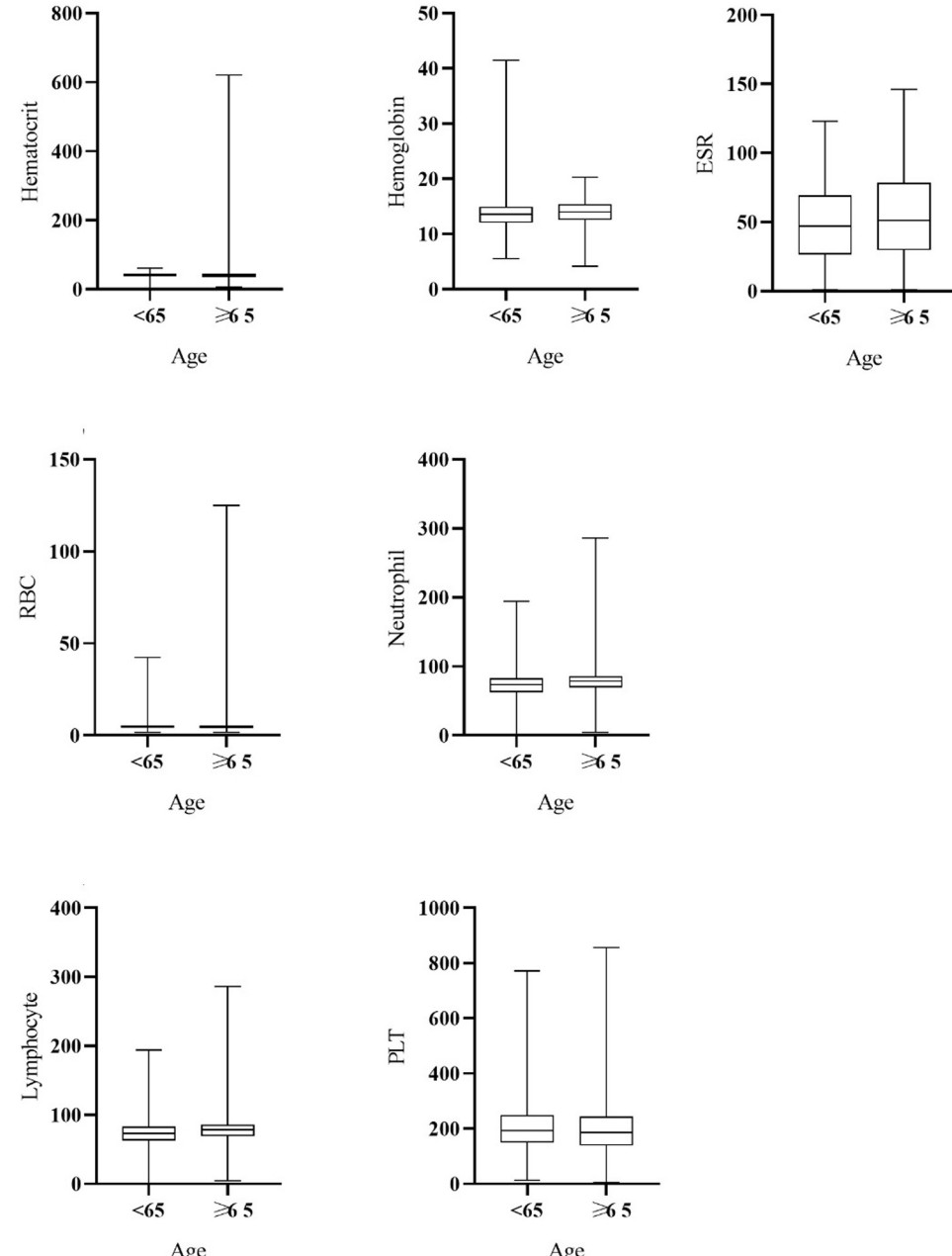

**Fig 4. Comparison of hematocrit, hemoglobin, red blood cell (RBC), neutrophil, lymphocyte, platelet (PLT), and erythrocyte sedimentation rate (ESR) level with age in patients with COVID-19.**

November 2021, encompassing the occurrence of five distinct pandemic waves associated with COVID-19 surge. Our study findings reveal significant variations in hematological parameters across different waves of the COVID-19 pandemic. Among the analyzed parameters, the highest increase in MCV and RDW-CV levels was observed during the first wave, substantial changes in lymphocyte count, platelet count, MCHC, and RDW-SD occurred during the second wave, notable alterations in WBC, RBC, neutrophil count, MCH, and PDW were observed during the third wave, and the most pronounced changes occurred in Hb, Hct, and ESR levels,

**Table 2. Comparison of hematological parameters based on gender among hospitalized patients during COVID-19 pandemic.**

| Parameters | Male (n = 816) | | | | Female (n = 685) | | | | P-Value |
|---|---|---|---|---|---|---|---|---|---|
| | Mean | SD | Median | IQR | Mean | SD | Median | IQR | |
| WBC | 7.47 | 5.32 | 6.3 | 4.15 | 7.49 | 8.38 | 6.2 | 4.5 | 0.33 |
| RBC | 5.32 | 5.4 | 4.94 | 0.9 | 4.74 | 2.64 | 4.58 | 0.72 | *<**0.001** |
| Neutrophil | 75.02 | 17.8 | 76.45 | 17.72 | 73.96 | 15.11 | 75.95 | 16.8 | 0.07 |
| Lymphocyte | 18.82 | 10.9 | 16.8 | 13.7 | 20.37 | 10.66 | 18.8 | 14.15 | ***0.002** |
| PLT | 192.6 | 88.6 | 179 | 100.5 | 218.62 | 93.45 | 202.5 | 90.5 | *<**0.001** |
| Hb | 14.33 | 2.59 | 14.5 | 2.68 | 13.11 | 2.147 | 13.1 | 2.1 | *<**0.001** |
| Hct | 42.89 | 21.29 | 42.75 | 9.98 | 39.61 | 5.25 | 39.85 | 5.9 | *<**0.001** |
| ESR | 49.5 | 28.81 | 48 | 45.5 | 54.20 | 29.36 | 51 | 46 | ***0.006** |

WBC: White blood cell, RBC: Red blood cell, PLT: Platelet, Hb: Hemoglobin, Hct: Hematocrit

ESR: Erythrocyte sedimentation rate.

Note: *P<0.01

during the fourth wave. Additionally, we observed that men had higher median levels of RBC, Hb, Hct, and ESR, while the mean levels of lymphocyte and platelet counts were lower in men compared to women. Moreover, a significant increase in mean neutrophil count was observed among deceased patients compared to those with full or partial recovery outcomes ($p < 0.001$).

Our study also found that older individuals and men had a higher susceptibility to COVID-19 infection, consistent with previous research [24, 25]. Among the different waves of the pandemic our observations revealed that the third wave had the most notable impact [26–28]. Our study emphasized that the third wave was characterized by significant changes in hematological parameters, including leukocytosis, neutrophilia, lymphopenia, increased RBC count, PLT count, Hb level, RDW-SD, and MCV. These findings underscore the distinct hematological alterations observed during the third wave of COVID-19.

Lymphocytes play a crucial role in the immune response against viral infections, and their quantification in CBC tests can serve as a valuable diagnostic tool for determining the nature of an infection and assessing the patient's clinical status [29, 30]. Lymphopenia can have a substantial impact on the adaptive immune responses of the host and significantly influence the

**Table 3. Comparison of hematological parameters based on in-hospital outcome among hospitalized patients during the COVID-19 pandemic.**

| Parameters | Full Recovery (n = 1220) | | | | Partial Recovery (n = 62) | | | | Death (n = 219) | | | | P-values |
|---|---|---|---|---|---|---|---|---|---|---|---|---|---|
| | Mean | SD | Median | IQR | Mean | SD | Median | IQR | Mean | SD | Median | IQR | |
| WBC | 7.18 | 3.61 | 6.25 | 5.35 | 7.15 | 7.41 | 6 | 3.8 | 8.58 | 5.13 | 7.2 | 5.8 | *<**0.001** |
| RBC | 4.74 | 0.718 | 4.68 | 1 | 5.09 | 4.65 | 4.76 | 0.85 | 4.93 | 3.96 | 4.65 | 0.98 | ***0.008** |
| Neutrophil | 74.27 | 13.18 | 75.1 | 17.5 | 73.02 | 16.45 | 74.6 | 17.1 | 80.97 | 12.9 | 83.15 | 12.32 | *<**0.001** |
| Lymphocyte | 19.11 | 9.76 | 19.35 | 13.3 | 20.86 | 10.78 | 19.5 | 13.7 | 13.44 | 9.04 | 11.6 | 9.65 | *<**0.001** |
| Hb | 13.6 | 1.978 | 13.4 | 2.22 | 13.84 | 2.43 | 13.9 | 2.6 | 13.5 | 2.73 | 13.4 | 3.2 | ***0.037** |
| Hct | 40.93 | 5.129 | 40.45 | 6.63 | 41.65 | 18.49 | 41.5 | 6.5 | 40.56 | 7.49 | 40.6 | 8.73 | 0.23 |
| PLT | 225.12 | 101.49 | 197.5 | 83.5 | 205.13 | 87.76 | 193 | 97 | 193.74 | 102.55 | 176.5 | 102 | ***0.007** |
| ESR | 49.12 | 25.16 | 49 | 41 | 50.93 | 29.19 | 48 | 47 | 55.96 | 29.24 | 56 | 44 | 0.06 |

WBC: White blood cell, RBC: Red blood cell, PLT: Platelet, Hb: Hemoglobin, Hct: Hematocrit

ESR: Erythrocyte sedimentation rate.

Note: *P<0.01

clinical progression of acute viral infections [31, 32]. While the normal levels of lymphocytes differ based on age, lymphopenia is commonly observed in the majority of severe cases of COVID-19, particularly in those who require hospitalization in intensive care units (ICUs) [29, 30, 33]. Furthermore, lymphopenia tends to be more severe in older adults, highlighting their increased vulnerability to the virus and the potential impact on their immune response [29]. Despite our findings demonstrated no significant correlation between lymphocyte count and age ($p = 0.8$), a significant difference in mean lymphocyte count was observed between deceased patients and those who achieved full or partial recovery ($p < 0.001$). This further emphasizes the importance of lymphocyte count as a potential prognostic marker in COVID-19, highlighting its association with disease severity and mortality risk. Previous studies have indicated that lymphopenia in COVID-19 is primarily associated with a decline in CD8+ T cells, while B cells and their numbers are relatively less affected [30, 34]. T-cell responses, especially those mediated by Th1 cells, have been recognized as critical for effective control of infections, including COVID-19 [35, 36]. T helper 1 (Th1) cells play a crucial role in coordinating immune responses against intracellular pathogens through the production of pro-inflammatory cytokines, activation of macrophages, and facilitation of the cytotoxic activity of CD8+ T cells. These cellular immune responses are essential for the eradication of infected cells and the resolution of viral infections [34]. Lymphopenia can result from various virus-related factors, including the occurrence of a cytokine storm characterized by elevated levels of the interleukin 6 (IL-6) [37].

Leukocytosis acts as a defensive response against viral infections, including COVID-19 [38, 39]. Our study observed the most significant increase in WBC count among patients affected during the 3rd wave. Notably, deceased patients exhibited higher WBC counts compared to those with partial and full recovery, indicating a significant association ($p < 0.001$). Similarly, Zhu et al. conducted a study that demonstrated higher WBC count in deceased patients, despite falling within the normal range. The authors suggested that the WBC count at the time of admission could serve as a predictive marker for patient deterioration and mortality [40]. Additionally, multiple studies have reported an association between leukocytosis and a poor prognosis in critically ill COVID-19 patients [38, 40, 41]. Finally, the current study did not observe a correlation between leukocytosis and age or gender variables; however, Pirsalehi et al. reported a significant association between leukocytosis and older adult men aged above 50 years [42].

Neutrophils-also known as granulocytes- are an integral part of the innate immune system that combats acute viral infections, including COVID-19, by mediating communication between the innate and acquired immune responses [43–45]. Studies have reported an initial increase of these cells in the epithelium of the nasopharynx, followed by migration to distant areas of the lung [43]. Patients with severe COVID-19 infection have exhibited alterations in the neutrophil count, as well as changes in their phenotype and function [43, 46]. Multiple studies have reported heightened gene expression of progenitor or immature neutrophils and increased levels of neutrophil-derived factors (such as RETN, HGF, and LCN2) in the plasma [47, 48]. These observations suggest a potential association between these markers and severe infection, as well as poor prognosis and increased mortality rates, contributing to our understanding of the underlying mechanisms involved in the progression and outcomes of severe infections. These findings also provide potential targets for therapeutic interventions and prognostic assessments [47]. Furthermore, severe inflammation and tissue damage resulting from COVID-19 infection are attributed to immune-pathogenesis responses, with neutrophils being one of the main players [49, 50]. Neutrophils contribute to tissue damage through the stimulation of cytokine, chemokine, and reactive oxygen species (ROS) production, as well as the release of neutrophil extracellular traps (NETs) [47, 51, 52]. All of these findings point to

an increase in neutrophils and their complications in critically ill COVID-19 patients [47]. In our study, we observed a significant increase in the mean neutrophil count among deceased patients compared to those with full or partial recovery. This finding aligns with existing literature and further supports the notion that elevated neutrophil count can serve as a valuable clinical marker of COVID-19 infection in the bloodstream.

RBCs play an important role in transporting oxygen from the lungs to the tissues and carrying carbon dioxide from the tissues back to the lungs, as well as maintaining biophysical consistency of the blood and the overall efficiency of the bloodstream [53]. COVID-19 infection has led researchers to focus on RBCs as potential targets of SARS-CoV-2 due to the occurrence of hypoxia and dyspnea [54, 55]. COVID-19 can bind to RBCs through the interaction with protein band-3, providing an alternative route for viral entry into host cells [47, 56]. This mode of infection, in addition to the respiratory organ infection through ACE2 receptors, highlights the potential involvement of RBCs in the pathogenesis of SARS-CoV-2 [47]. SARS-CoV-2 binding to RBCs may not directly facilitate virus replication, but it has notable consequences on various RBC functions, including the release of oxygen from RBCs [57, 58]. The interaction between SARS-CoV-2 and RBCs has the potential to disrupt their normal oxygen-carrying capacity, thereby leading to impaired oxygen delivery to tissues and contributing to the development of hypoxia in individuals with COVID-19 [59]. Furthermore, changes in the levels of oxidized glutathione and key enzymes involved in countering oxidative stress, such as superoxide dismutase 1 (SOD1), glucose-6-phosphate dehydrogenase (G6PD), and peroxiredoxin (Prxs), result in heightened susceptibility of red blood cells to reactive oxygen species. This elevated oxidative stress may lead to cell lysis and compromise the efficient transport of oxygen by RBCs [60]. Significant research on Hb, a crucial component within RBCs, has demonstrated its interaction with various components of SARS-CoV-2, including ORF1ab, ORF3a, ORF7a, ORF8a, and ORF10. These interactions can lead to denaturation of Hb and a subsequent reduction in its quantity [59, 61]. The described phenomenon exerts a significant impact on oxygen transport reduction, leading to hypoxia and the manifestation of a multifaceted syndrome, a prominent feature observed in COVID-19 cases. Numerous studies conducted on critically ill COVID-19 patients have consistently reported a decrease in Hb levels, which may serve as a potential indicator of a poor prognosis in individuals requiring mechanical ventilation [59, 62]. Consistent with these findings, our study also demonstrated a reduction in Hb levels among deceased patients. Additionally, a statistically significant difference was observed between the Hb levels of deceased patients and those who had recovered from the infection, underscoring the potential prognostic value of Hb in assessing the clinical outcome of COVID-19 patients.

## 5. Conclusion

Our study observed distinct patterns in hematological parameters across different waves of the COVID-19 pandemic. Specifically, the 1st wave exhibited the highest increase in MCV and RDW-CV levels, while the 2nd wave showed significant changes in lymphocyte count, MCHC, PLT count, and RDW-SD. In the 3rd wave, notable alterations were observed in WBC, RBC, neutrophil count, MCH, and PDW, and the 4th wave had the most pronounced changes in Hb, Hct, and ESR levels. These findings indicate that hematological parameters can serve as valuable predictive biomarkers for assessing disease status and clinical outcomes in each wave of the COVID-19 pandemic. Healthcare professionals can gain valuable insights into the progression and prognosis of COVID-19 cases by taking these parameters into account, enabling more informed decision-making and personalized patient management strategies.

## Supporting information

**S1 File.**
(DOCX)

## Author Contributions

**Data curation:** Faezeh Afkhami Aghda.

**Formal analysis:** Shahriar Dabiri.

**Investigation:** Sara Pourshaikhali.

**Methodology:** Javad Charostad.

**Project administration:** Javad Charostad, Mohsen Nakhaie.

**Resources:** Yaser ghelmani.

**Supervision:** Mohsen Nakhaie.

**Writing – original draft:** Javad Charostad, Azam dehghani, Akram Astani, Ehsan Kakavand.

**Writing – review & editing:** Mohammad Rezaei Zadeh Rukerd, Pouria Pourzand, Mohsen Nakhaie.

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
