## [Decision Letter · Decision Letter 0]

3 Jan 2023

PONE-D-22-21644Evaluation of Hematological Parameters Alterations in Different Waves of COVID-19 PandemicPLOS ONE

Dear Dr. Nakhaie,

Thank you for submitting your manuscript to PLOS ONE. After careful consideration, we feel that it has merit but does not fully meet PLOS ONE’s publication criteria as it currently stands. Therefore, we invite you to submit a revised version of the manuscript that addresses the points raised during the review process.

We look forward to receiving your revised manuscript.

Kind regards,

Bhaswati Chatterjee, PhD

Academic Editor

PLOS ONE

and https://journals.plos.org/plosone/s/file?id=ba62/PLOSOne_formatting_sample_title_authors_affiliations.pdf.

2. Thank you for including your ethics statement:  "N/A".  

(1). For studies reporting research involving human participants, PLOS ONE requires authors to confirm that this specific study was reviewed and approved by an institutional review board (ethics committee) before the study began. Please provide the specific name of the ethics committee/IRB that approved your study, or explain why you did not seek approval in this case.

(2). Please provide additional details regarding participant consent. In the ethics statement in the Methods and online submission information, please ensure that you have specified (1) whether consent was informed and (2) what type you obtained (for instance, written or verbal, and if verbal, how it was documented and witnessed). If your study included minors, state whether you obtained consent from parents or guardians. If the need for consent was waived by the ethics committee, please include this information.

3. PLOS requires an ORCID iD for the corresponding author in Editorial Manager on papers submitted after December 6th, 2016. Please ensure that you have an ORCID iD and that it is validated in Editorial Manager. To do this, go to ‘Update my Information’ (in the upper left-hand corner of the main menu), and click on the Fetch/Validate link next to the ORCID field. This will take you to the ORCID site and allow you to create a new iD or authenticate a pre-existing iD in Editorial Manager. Please see the following video for instructions on linking an ORCID iD to your Editorial Manager account: https://www.youtube.com/watch?v=_xcclfuvtxQ.

4. Please amend your authorship list in your manuscript file to include authors list.

Reviewers' comments:

Reviewer's Responses to Questions

**Comments to the Author**

1. Is the manuscript technically sound, and do the data support the conclusions?

Reviewer #1: Partly

Reviewer #2: Partly

Reviewer #3: Yes

2. Has the statistical analysis been performed appropriately and rigorously? 

Reviewer #1: No

Reviewer #2: Yes

Reviewer #3: Yes

3. Have the authors made all data underlying the findings in their manuscript fully available?

Reviewer #1: Yes

Reviewer #2: No

Reviewer #3: Yes

4. Is the manuscript presented in an intelligible fashion and written in standard English?

Reviewer #1: No

Reviewer #2: Yes

Reviewer #3: No

5. Review Comments to the Author

Reviewer #1: This paper looks at hematological parameters in different waves, by recruiting patients at the hospital. By definition these people are not healthy and a control (COVID negative) is required at each wave.

We have no information on the waves (viral strain, evaluated infection levels in the general population) that renders the information un-informative.

These type of studies need multiparametric studies. THere are too many confusing factors (not only age and sex).

Reviewer #2: 20 December 2022

PLoS One

ISSN: 1932-6203

RE: Invitation to review manuscript number PONE-D-22-21644 entitled “Evaluation of Hematological Parameters Alterations in Different Waves of COVID-19 Pandemic [sic]”

Dear Editors,

I would like to thank your for the opportunity to review a manuscript on behalf of PLoS One. In this cross-sectional study, the authors describe routine hematological parameters associated with diseases severity among COVID-19 patients admitted to Shahid Sadoughi hospital (Yazd, Iran). While this is not the first study to describe the association of hematological parameters and COVID-19 severity, the findings presented by the authors add to the external validity of using this approach do determine patient prognosis. The manuscript is succinct and technically sound, but requires significant revision to be considered for publication in PLoS One. I would encourage the authors to revise their manuscript since their findings could be helpful for future systematic reviews and meta-analysis such as Asghar et al. 2020 JCHIMP 10(6): 508-513.

1. On occasion, spelling and verbiage is used inconsistently. I would encourage the authors to revise their manuscript for consistency. For example,

- While covid-19 mainly affects the respiratory system […], So far several models for the prognosis of the disease of COVID-19 […], Numerous reports of the disorder of hematological parameters levels in Covid-19 patients […]

- […] the body is another target of this wild virus. It is unclear what the authors signify by “wild virus”.

- Some words are unnecessarily capitalized (hematological Parameters; and There is an urgent; And Also, from the early detection; Quantitative Characteristics of patients)

2. Some statements made by the authors should be substantiated with references. I would encourage the authors to review the current literature to better position their own study and findings. For example,

- So far, several models for the prognosis for the disease of COVID-19 have been reported. What are these models?

- A study conducted in five different countries in Asia, Africa, and the United States shows that CBB is the most common laboratory test. These studies should be cited.

- Numerous reports of the disorder of hematological parameters levels in Covid-19 patients have been mentioned in various studies, some of which had a significant relationship with the clinical status and prognosis of the patients. It’s unclear which studies are being referenced here.

- Previous studies showed that leukocytosis predicts a poor prognosis in critically ill Covid-19 patients. Which studies?

3. The authors should provide a justification for their sample size. Why did they only select n=300 patients per wave of COVID-19? The authors potentially have access to a much larger dataset and this could strengthen their findings. Also, the authors should provide a justification as to why they chose not to include asymptomatic controls/negative for comparison.

4. In general, how much time was there between a positive RT-PCR test and hematological assessment in the patient population? Was this similar in all three categories of patients (i.e., partially recovered, recovered, and deceased)? How would delays between a positive RT-PCR and hematological assessment affect the study’s findings? Sampling delays could significantly bias the study’s findings and should be acknowledged by the authors.

5. Some acronyms are not defined (ex: MCH, MCHC, MCV, PDW, PLT). While these acronyms might be common in hematology, PLoS One’s broad readership would benefit from additional definitions.

6. The distribution of patients aged < and ≥ 65 years […]. Is there a value missing after the “<”?

7. The authors should provide a definition for “partial recovery” and “recovery”. The difference between these two categories is not clear.

8. It is difficult to appreciate the study’s findings because continuous data is being presented in a Table versus Figures. I would highly recommend that the authors use Figures present the data found in Tables 2-5. Currently, it is difficult to appreciate changes in hematological parameters from the 1st to 5th waves of COVID-19, age groups, sex, and disease severity. In addition, it is not very clear from the tables where these significant changes are happening. I would recommend the authors use box-and-whisker plots (where applicable) to present their findings. Using figures would also complement the text.

9. As stated by the authors, the findings from this study could be useful for optimizing clinical decisions in countries that suffer from significant shortages of medical resources, potentially reducing mortality rates. The authors show that hematological parameters vary from wave to wave of COVID-19. However, they do not discuss how these changes affect the association of hematological parameters and disease severity. Do we see a change in sensitivity and specificity from wave to wave? The authors should report sensitivity and specificity data to (1) allow comparisons with other similar studies and (2) allow the reader to assess the usefulness of using hematological parameters for disease prognosis.

10. The authors specify that “those patients who had missing data in their hematological parameters […] were negatively excluded” but the N= reported in each table would suggest otherwise. For example, in Table 3 N=716 values are reported for hematocrit while N=687 values are being reported for lymphocyte. Shouldn’t each parameter have an equal number of patients? Table 1 suggests that 717 patients <65 were included in this study, but Table 3 reports 716 hematocrit measurements, 666 neutrophil measurements, etc. Some further explanation would be valuable here. The authors should report the sample size (n=), denominators (N=), and proportions (%) for each hematological parameter in each Table to facilitate interpretation. A flow-chart could also assist the reader in understanding how patients were selected, included, excluded, and which hematological parameters were available at the time of the study.

Reviewer #3: The manuscript entitled “Evaluation of Hematological Parameters Alterations in Different Waves of COVID-19 Pandemic” described hematological parameters during Covid-19 infections in Yazd city in Iran. Although molecular testing by RT-PCR is already a good tool for Covid-19 diagnosis, I believe, this together with the authors that hematological changes together with PCR can help to better predict the outcome of Covid-19 patients. I think therefore, and despite the relatively poor quality of English of this manuscript, that this study is an important addition to the literature of Covid-19 in Iran and in the world. However, I have to suggest that the overall scientific quality be improved.

Major points

1. I suggest the authors should add a quick description of the study city or site for international readers

2. Throughout the main text, the authors emphasized on different waves of Covid-19 in Iran but it is not clear for the reader when and for how long these waves occurred. I therefore recommend the authors to present a figure showing different waves of Covid-19 in Iran.

4. There are a number of grammatical mistakes and abbreviations in the text without any initial explanations during the first appearance. Authors should correct that.

5. Some unnecessarily long sentences should also be broken into at least two.

6. PLOS authors have the option to publish the peer review history of their article (what does this mean?). If published, this will include your full peer review and any attached files.

Reviewer #1: No

Reviewer #2: No

Reviewer #3: **Yes: **Demanou Maurice

---

## [Author Response · Author response to Decision Letter 0]

17 Mar 2023

We greatly appreciate the effort and time you have dedicated to providing valuable feedback on our manuscript. We are grateful to the reviewers for their insightful comments on my paper. We have been able to incorporate changes to reflect most of the suggestions provided by the reviewers.

Reviewer #1:

This paper looks at hematological parameters in different waves, by recruiting patients at the hospital. By definition these people are not healthy and a control (COVID negative) is required at each wave.

Response: thanks for your valuable suggestion. The current research was performed in the main COVID-19 center hospital, Shahid sadoughi hospital, affiliated with Yazd University of Medical Sciences, in Yazd province, that at that time, the focus was on admitting people with COVID-19 and all the departments of the hospital were assigned to this positive group, so unfortunately non-covid-19 people were not admitted to this hospital.

We have no information on the waves (viral strain, evaluated infection levels in the general population) that renders the information un-informative.

Response: Considering that these data were collected from the city of Yazd and reached us, except in the third wave, which was the type of virus variant x, more information about the type of virus variant in the rest of the waves and the level of involvement of the general Population in the form of documents not available.

These type of studies need multiparametric studies. There are too many confusing factors (not only age and sex).

Response: Thank you for your valuable point. Since the data collection time was at the peak of the COVID-19 and at that time our data source was the files extracted from the Hospital information system, the variable items and existing confusing factors were limited for the entire study population.

Reviewer #2: 

1. On occasion, spelling and verbiage is used inconsistently. I would encourage the authors to revise their manuscript for consistency. For example,

- While covid-19 mainly affects the respiratory system […], So far several models for the prognosis of the disease of COVID-19 […], Numerous reports of the disorder of hematological parameters levels in Covid-19 patients […]

- […] the body is another target of this wild virus. It is unclear what the authors signify by “wild virus”.

- Some words are unnecessarily capitalized (hematological Parameters; and There is an urgent; And Also, from the early detection; Quantitative Characteristics of patients)

Response: Thank you for your valuable point. All the items that were correctly mentioned have been corrected.

2. Some statements made by the authors should be substantiated with references. I would encourage the authors to review the current literature to better position their own study and findings. For example,

- So far, several models for the prognosis for the disease of COVID-19 have been reported. What are these models?

Response: According to PMID: 32265220 study, various models have been made for this prediction, which was added to the text references at your suggestion.

- A study conducted in five different countries in Asia, Africa, and the United States shows that CBB is the most common laboratory test. These studies should be cited.

Response: According to PMID: 30535132 study, which was added to the text references at your suggestion.

- Numerous reports of the disorder of hematological parameters levels in Covid-19 patients have been mentioned in various studies, some of which had a significant relationship with the clinical status and prognosis of the patients. It’s unclear which studies are being referenced here.

Response: According to PMID: 30535132 study, which was added to the text references at your suggestion.

- Previous studies showed that leukocytosis predicts a poor prognosis in critically ill Covid-19 patients. Which studies?

Response: According to PMID: 32822430, PMID: 32533986, and PMID: 34126954 studies, several studies showed that leukocytosis predicts a poor prognosis, which were added to the text references at your suggestion.

3. The authors should provide a justification for their sample size. Why did they only select n=300 patients per wave of COVID-19? The authors potentially have access to a much larger dataset and this could strengthen their findings. Also, the authors should provide a justification as to why they chose not to include asymptomatic controls/negative for comparison.

Response: G-Power software was used to calculate the sample size. Considering the effect size of 0.09, the alpha error of 0.05, the power of the test 0.80, and the number of groups equal to 5, the sample size was calculated as 1480, as a result, approximately 300 samples were needed for each group.

Below you can see the output of this software along with the alpha and beta error graph.

F tests - ANOVA: Fixed effects, omnibus, one-way

Analysis: A priori: Compute required sample size 

Input: Effect size f = 0.09

 α err prob = 0.05

 Power (1-β err prob) = 0.80

 Number of groups = 5

Output: Noncentrality parameter λ = 11.9880000

 Critical F = 2.3779612

 Numerator df = 4

 Denominator df = 1475

 Total sample size = 1480

4. In general, how much time was there between a positive RT-PCR test and hematological assessment in the patient population? Was this similar in all three categories of patients (i.e., partially recovered, recovered, and deceased)? How would delays between a positive RT-PCR and hematological assessment affect the study’s findings? Sampling delays could significantly bias the study’s findings and should be acknowledged by the authors.

Response: Thanks for your appropriate point. All PCR tests were taken for all people (Referred individuals with symptoms) in all groups at the same time and upon their arrival at the hospital. The cases with PCR positive result (Up to 24 hours) were subjected to the evaluation of hematological parameters. This item mentioned in the manuscript “Study design and participants” and highlighted. 

5. Some acronyms are not defined (ex: MCH, MCHC, MCV, PDW, PLT). While these acronyms might be common in hematology, PLoS One’s broad readership would benefit from additional definitions.

Response: Thank you for your valuable point. All the items that were correctly mentioned have been corrected.

6. The distribution of patients aged < and ≥ 65 years […]. Is there a value missing after the “<”?

Response: Thank you for your valuable point. All the items that were correctly mentioned have been corrected.

7. The authors should provide a definition for “partial recovery” and “recovery”. The difference between these two categories is not clear.

Response: Partial recovery refers to people who, although the process of improving their condition was observed, but they had not fully recovered and they needed to continue the treatment, so that either anti-COVID-19 medication should be received at home or, if necessary, they would go to the doctor's office or hospital to follow up their condition. Recovery refers to patients were those who were completely treated and did not need to go to the doctor or hospital again.

8. It is difficult to appreciate the study’s findings because continuous data is being presented in a Table versus Figures. I would highly recommend that the authors use Figures present the data found in Tables 2-5. Currently, it is difficult to appreciate changes in hematological parameters from the 1st to 5th waves of COVID-19, age groups, sex, and disease severity. In addition, it is not very clear from the tables where these significant changes are happening. I would recommend the authors use box-and-whisker plots (where applicable) to present their findings. Using figures would also complement the text.

Response: Thank you for your valuable point. All the items that were correctly mentioned have been corrected and revealed in box-and-whisker plot.

9. As stated by the authors, the findings from this study could be useful for optimizing clinical decisions in countries that suffer from significant shortages of medical resources, potentially reducing mortality rates. The authors show that hematological parameters vary from wave to wave of COVID-19. However, they do not discuss how these changes affect the association of hematological parameters and disease severity. Do we see a change in sensitivity and specificity from wave to wave? The authors should report sensitivity and specificity data to (1) allow comparisons with other similar studies and (2) allow the reader to assess the usefulness of using hematological parameters for disease prognosis.

Response: If we have understood your meaning correctly, our answer to you is: you ask a good question, as the science of epidemiology shows us, the prevalence of a disease has an effect on the sensitivity and specificity of its diagnostic test. However, in this study, we did not seek to investigate the prevalence and only studied the patients with covid-19, who were already confirmed to be sick with a gold standard (PCR).

10. The authors specify that “those patients who had missing data in their hematological parameters […] were negatively excluded” but the N= reported in each table would suggest otherwise. For example, in Table 3 N=716 values are reported for hematocrit while N=687 values are being reported for lymphocyte. Shouldn’t each parameter have an equal number of patients? Table 1 suggests that 717 patients <65 were included in this study, but Table 3 reports 716 hematocrit measurements, 666 neutrophil measurements, etc. Some further explanation would be valuable here. The authors should report the sample size (n=), denominators (N=), and proportions (%) for each hematological parameter in each Table to facilitate interpretation. A flow-chart could also assist the reader in understanding how patients were selected, included, excluded, and which hematological parameters were available at the time of the study.

Response: Thank you for your valuable point. All the items that were correctly mentioned have been corrected in tables. Due to the limitations of the laboratory and personnel workload, all tests are not performed routinely in this center, and the reason for the difference between the level of lymphocytes and the hematocrit is due to this.

Reviewer #3: 

1. I suggest the authors should add a quick description of the study city or site for international readers

Response: Thank you for your valuable point. All the items that were correctly mentioned have been corrected.

2. Throughout the main text, the authors emphasized on different waves of Covid-19 in Iran but it is not clear for the reader when and for how long these waves occurred. I therefore recommend the authors to present a figure showing different waves of Covid-19 in Iran.

Response: Thank you for your valuable point. All the items that were correctly mentioned have been corrected and mentioned in figure 2.

3. There are a number of grammatical mistakes and abbreviations in the text without any initial explanations during the first appearance. Authors should correct that.

Response: Thank you for your valuable point. All the items that were correctly mentioned have been corrected.

4. Some unnecessarily long sentences should also be broken into at least two.

Response: Thank you for your valuable point. All the items that were correctly mentioned have been corrected.

---

## [Decision Letter · Decision Letter 1]

17 Apr 2023

PONE-D-22-21644R1Evaluation of Hematological Parameters Alterations in Different Waves of COVID-19 Pandemic: A Cross-Sectional StudyPLOS ONE

Dear Dr. Nakhaie,

Thank you for submitting your manuscript to PLOS ONE. After careful consideration, we feel that it has merit but does not fully meet PLOS ONE’s publication criteria as it currently stands. Therefore, we invite you to submit a revised version of the manuscript that addresses the points raised during the review process.

We look forward to receiving your revised manuscript.

Kind regards,

Bhaswati Chatterjee, PhD

Academic Editor

PLOS ONE

Reviewers' comments:

Reviewer's Responses to Questions

**Comments to the Author**

1. If the authors have adequately addressed your comments raised in a previous round of review and you feel that this manuscript is now acceptable for publication, you may indicate that here to bypass the “Comments to the Author” section, enter your conflict of interest statement in the “Confidential to Editor” section, and submit your "Accept" recommendation.

Reviewer #1: (No Response)

Reviewer #2: (No Response)

Reviewer #3: (No Response)

2. Is the manuscript technically sound, and do the data support the conclusions?

Reviewer #1: No

Reviewer #2: Yes

Reviewer #3: Yes

3. Has the statistical analysis been performed appropriately and rigorously? 

Reviewer #1: I Don't Know

Reviewer #2: No

Reviewer #3: Yes

4. Have the authors made all data underlying the findings in their manuscript fully available?

Reviewer #1: No

Reviewer #2: No

Reviewer #3: Yes

5. Is the manuscript presented in an intelligible fashion and written in standard English?

Reviewer #1: No

Reviewer #2: Yes

Reviewer #3: Yes

6. Review Comments to the Author

Reviewer #1: The authors did not respond to any of my comments.

My comments remain, and I believe that the data provided is not sufficient for publication.

Reviewer #2: While the current manuscript is a significant improvement compared to the previous version, it requires additional revision to be considered for publication in PLoS ONE in my personal opinion.

1. The authors still have not provided a justification as to why they chose not to include asymptomatic/negative/healthy controls for comparison.

2. The manuscript is missing several references. For example:

[…] and their reduction can be extremely harmful to infection such as COVID-19 (ref).

[…] and their decreased number are less affected by infection (ref).

I would encourage the authors to carefully revise their manuscript for accuracy and completeness.

3. Several typographical errors are present in the manuscript, including in the figures. I would encourage the authors to carefully refine their manuscript. For example:

Figure 3, X-axis, “RDW_CV”, 2th (should be 2nd), 3th (should be 3rd), 4th and 5th

Table 1. […] with COVID-19 during the covid-19 pandemic

4. The authors note in the results that there was a significant association between hematological parameters amongst “younger” and “elderly” patients. More specifically, the mean of WBC, neutrophil, and ESR were significantly higher in the “elderly” than in the “younger ones”. The level of hematocrit, hemoglobin, RBC, and ESR parameters were also significantly higher in men than in women, while the median level of lymphocyte and platelets were lower in men versus women. In addition, the distribution of patient age varies between the different waves of the COVID-19 pandemic (Table 1). However, aggregate data is presented in Figure 3. A stratified analysis based on different age categories would be helpful here. It is unclear if the differences observed are due to the different SARS-CoV-2 variants or the changing mean age of the patient population from wave-to-wave.

Reviewer #3: I'd that the authors for their effort to address the reviewers' comments to improve the overall quality of their manuscript.

The figure 2 showing the time frame of each of the covid-19 waves in Iran and its relationship with the patients outcome is particularly appreciated.

However, I still believe that one of my major comments below was not addressed.

Major points

1. I suggest the authors should add a quick description of the study city or site for international readers

This is important for international reader to better understand the epidemiological pattern of COVID-19 in the city of Yazd.

I recommend the authors to add a study site description section together with a map of Iran for more illustration.

7. PLOS authors have the option to publish the peer review history of their article (what does this mean?). If published, this will include your full peer review and any attached files.

Reviewer #1: No

Reviewer #2: No

Reviewer #3: **Yes: **Maurice Demanou

---

## [Author Response · Author response to Decision Letter 1]

27 May 2023

We deeply value the dedication and time you have invested in offering valuable feedback on our manuscript. We extend our sincere gratitude to the reviewers for their perceptive comments on our paper. We have successfully integrated modifications to address the majority of the suggestions put forth by the reviewers.

Reviewer #1: 

Q: The authors did not respond to any of my comments.

My comments remain, and I believe that the data provided is not sufficient for publication.

A: Thank you for your comment. We have made sincere efforts to incorporate the changes based on your feedback, and we greatly appreciate your detailed comments. However, it is important to acknowledge that some limitations exist within our study, and we have addressed these limitations to the best of our abilities. The research which conducted at Shahid Sadoughi Hospital, the main COVID-19 center affiliated with Yazd University of Medical Sciences in Yazd province. The current study employed a cross-sectional approach, as evident from the main title of the manuscript. The primary aim of the study was to offer a comprehensive overview of different hematological parameters in individuals who tested positive for COVID-19 throughout various waves, ranging from the first wave to the fifth wave. The main emphasis was placed on describing these parameters and presenting the findings obtained from the descriptive analysis conducted. Additionally, due to the limitations of routine hospital tests in detecting the specific variant type of the SARS-CoV-2 virus, the determination of variant types for each wave in our study was carried out by the Ministry of Health in our country. In order to accurately identify the variant, sequencing is required, which was not performed during the analysis to ensure the accuracy of the results. Therefore, the analyses conducted in our study were not based on the specific variant types. The data collection process occurred during the peak of the COVID-19 pandemic, where our primary data source consisted of files extracted from the Hospital Information System. Consequently, the study was limited by the availability of variable items and other confounding factors for the entire study population.

Reviewer #2: While the current manuscript is a significant improvement compared to the previous version, it requires additional revision to be considered for publication in PLoS ONE in my personal opinion.

Q: The authors still have not provided a justification as to why they chose not to include asymptomatic/negative/healthy controls for comparison.

A: Thank you for your valuable comment. The present study utilized a cross-sectional approach, as indicated in the main title of the manuscript. Our primary objective was to provide a general overview of various hematological parameters specifically among COVID-19-positive patients across different waves (from the first to the fifth wave). The main focus was on describing these parameters and reporting the findings resulting from this descriptive analysis.

Q: The manuscript is missing several references. For example:

[…] and their reduction can be extremely harmful to infection such as COVID-19 (ref).

[…] and their decreased number are less affected by infection (ref).

I would encourage the authors to carefully revise their manuscript for accuracy and completeness.

A: Thank you very much for your feedback. We appreciate your acknowledgement that our comment has enhanced the value of the manuscript. We have taken great care in revising and correcting all the mentioned items accordingly.

3. Several typographical errors are present in the manuscript, including in the figures. I would encourage the authors to carefully refine their manuscript. For example:

Figure 3, X-axis, “RDW_CV”, 2th (should be 2nd), 3th (should be 3rd), 4th and 5th

Table 1. […] with COVID-19 during the covid-19 pandemic

A: Thank you very much for your feedback. We appreciate your acknowledgement that our comment has enhanced the value of the manuscript. We have taken great care in revising and correcting all the mentioned items accordingly.

Q: The authors note in the results that there was a significant association between hematological parameters amongst “younger” and “elderly” patients. More specifically, the mean of WBC, neutrophil, and ESR were significantly higher in the “elderly” than in the “younger ones”. The level of hematocrit, hemoglobin, RBC, and ESR parameters were also significantly higher in men than in women, while the median level of lymphocyte and platelets were lower in men versus women. In addition, the distribution of patient age varies between the different waves of the COVID-19 pandemic (Table 1). However, aggregate data is presented in Figure 3. A stratified analysis based on different age categories would be helpful here. It is unclear if the differences observed are due to the different SARS-CoV-2 variants or the changing mean age of the patient population from wave-to-wave.

A: Thank you for your response. We appreciate your recognition of the value our comment has added to the manuscript. We understand that our study encountered limitations, one of which was the unavailability of variant strain information for individual waves. Consequently, the study focused solely on investigating age and gender, and the reported results are based on these variables. It is important to acknowledge and address these limitations within the study.

Reviewer #3: I'd that the authors for their effort to address the reviewers' comments to improve the overall quality of their manuscript.

The figure 2 showing the time frame of each of the covid-19 waves in Iran and its relationship with the patients outcome is particularly appreciated.

However, I still believe that one of my major comments below was not addressed.

Major points

1. I suggest the authors should add a quick description of the study city or site for international readers

This is important for international reader to better understand the epidemiological pattern of COVID-19 in the city of Yazd.

I recommend the authors to add a study site description section together with a map of Iran for more illustration.

A: Thank you very much for your feedback. We appreciate your acknowledgement that our comment has enhanced the value of the manuscript. We have taken great care in revising and correcting all the mentioned items accordingly. Additionally, a map form Yazd city has been added as a supplementary figure. Moreover, this paragraph has been added to article: 

“Yazd city, is the capital of Yazd province the Central geographic region in Iran, and is recognized as a World Heritage Site by UNESCO [1]. Yazd has a hot desert climate and is known as the driest major city in Iran. Yearly precipitation amount in this town is less than 50 millimeters with summer temperatures very frequently above 40°C (104°F) in blazing sunshine with no humidity. Shahid Sadoughi University of Medical Sciences (SSU) is the only public medical university in this province and Shahid Sadoughi Hospital is the main affiliated hospital with SSU.” 

1. Rezaee M, Charrahi Z. Spatial Planning and Tourism Development with Sustainability Model of the Territorial Tourist with Land Use Approach. Int J Soc Bus Sci. 2020;14: 1007–1012.

---

## [Decision Letter · Decision Letter 2]

12 Jul 2023

PONE-D-22-21644R2Evaluation of Hematological Parameters Alterations in Different Waves of COVID-19 Pandemic: A Cross-Sectional StudyPLOS ONE

Dear Dr. Nakhaie,

Thank you for submitting your manuscript to PLOS ONE. After careful consideration, we feel that it has merit but does not fully meet PLOS ONE’s publication criteria as it currently stands. Therefore, we invite you to submit a revised version of the manuscript that addresses the points raised during the review process.

We look forward to receiving your revised manuscript.

Kind regards,

Bhaswati Chatterjee, PhD

Academic Editor

PLOS ONE

Journal Requirements:

Reviewers' comments:

Reviewer's Responses to Questions

**Comments to the Author**

1. If the authors have adequately addressed your comments raised in a previous round of review and you feel that this manuscript is now acceptable for publication, you may indicate that here to bypass the “Comments to the Author” section, enter your conflict of interest statement in the “Confidential to Editor” section, and submit your "Accept" recommendation.

Reviewer #2: (No Response)

Reviewer #3: All comments have been addressed

2. Is the manuscript technically sound, and do the data support the conclusions?

Reviewer #2: No

Reviewer #3: Yes

3. Has the statistical analysis been performed appropriately and rigorously? 

Reviewer #2: Yes

Reviewer #3: Yes

4. Have the authors made all data underlying the findings in their manuscript fully available?

Reviewer #2: No

Reviewer #3: Yes

5. Is the manuscript presented in an intelligible fashion and written in standard English?

Reviewer #2: Yes

Reviewer #3: Yes

6. Review Comments to the Author

Reviewer #2: I commend the authors for their latest revision of the manuscript. Although significant improvements have been made since its initial submission, in my personal opinion, further revisions are needed for it to be considered for publication in PLoS ONE.

1. Typographical errors are still prevalent in the manuscript, including within the figures and tables. I would encourage the authors to revise their manuscript carefully.

2. I strongly encourage the authors to reconsider the conclusions they have drawn from their statistical analysis. Specifically, “Among the different waves of the pandemic, we observed that the third wave, associated with the B.1.1.413 variant originating from Western countries, Australia, and Canada, had the most significant impact in our observations”. As mentioned by the authors, mean ages and outcomes (partial recovery, full recovery, and death) varied significantly from wave to wave, therefore it is difficult to determine whether changes in hematological parameters were a result of the “third wave” or simply due to an older patient population and/or a higher proportion of deceased patients, without further statistical analysis.

- To the best of my understanding, the authors’ objective was to describe cohorts of COVID-19 patients using age, sex, disease outcome and hematological parameters during multiple waves of the pandemic. If this is the case, the current statistical analysis is appropriate. However, the authors should reconsider their phrasing to avoid making associations between specific waves and changes in hematological parameters throughout the manuscript.

- If the authors would like to make associations between their observations and specific waves of COVID-19, they should consider undertaking a matched analysis based on age, sex, disease outcome, and the severity of symptoms at time of hospital admission (if available). By matching patients with similar characteristics, the authors could control for potential confounding factors and improve the robustness of their analysis.

Reviewer #3: (No Response)

7. PLOS authors have the option to publish the peer review history of their article (what does this mean?). If published, this will include your full peer review and any attached files.

Reviewer #2: No

Reviewer #3: **Yes: **DEMANOU Maurice

---

## [Author Response · Author response to Decision Letter 2]

21 Jul 2023

We greatly appreciate the effort and time you have dedicated to providing valuable feedback on our manuscript. We are grateful to the reviewers for their insightful comments on my paper. We have been able to incorporate changes to reflect most of the suggestions provided by the reviewers.

Reviewer #2:

Q: Typographical errors are still prevalent in the manuscript, including within the figures and tables. I would encourage the authors to revise their manuscript carefully.

A: We sincerely appreciate the time and effort you dedicated to reviewing our manuscript. Your insightful comments and constructive feedback have been immensely valuable in improving the quality of our research. We would like to express our gratitude specifically for pointing out the typographical errors present in the original version of the manuscript, including those within the figures and tables. We have taken your feedback seriously and conducted a thorough revision to address these issues diligently. During the revision process, we carefully corrected all the typographical errors in the main text, figures, and tables. Additionally, we have highlighted all the changes made to facilitate a clear understanding of the revisions. Moreover, based on the feedback received and our latest data, we have revised the content accordingly, aiming to enhance the accuracy and clarity of our findings. We hope that our efforts in revising the manuscript have met your expectations. Should you require any further information or have any additional concerns, we would be more than happy to address them promptly.

Q: I strongly encourage the authors to reconsider the conclusions they have drawn from their statistical analysis. Specifically, “Among the different waves of the pandemic, we observed that the third wave, associated with the B.1.1.413 variant originating from Western countries, Australia, and Canada, had the most significant impact in our observations”. As mentioned by the authors, mean ages and outcomes (partial recovery, full recovery, and death) varied significantly from wave to wave, therefore it is difficult to determine whether changes in hematological parameters were a result of the “third wave” or simply due to an older patient population and/or a higher proportion of deceased patients, without further statistical analysis.

A: We acknowledge and understand your concern regarding the conclusions drawn from our statistical analysis, particularly in regard to the impact of the B.1.1.413 variant and the higher mortality observed in the third wave of the pandemic. Your valuable comment prompted us to reevaluate our approach, and we wholeheartedly agree with your suggestion to refrain from attributing causation without conducting further statistical analysis. Upon careful consideration, we have made significant revisions to our manuscript to better clarify our findings. Specifically, we have removed statements implying a causal relationship between the B.1.1.413 variant and higher mortality during the third wave. Instead, we now emphasize that our epidemiological analysis revealed a higher mortality rate during this wave, and the median age of patients was indeed higher in the third wave compared to previous ones. As you astutely pointed out, variations in mean ages and outcomes among the waves introduce potential confounding factors that could influence the observed changes in hematological parameters. Therefore, we have reframed our discussion to acknowledge these limitations and have avoided drawing definitive conclusions regarding the cause of the observed differences in mortality rates. Instead, we have opted for a more cautious and data-driven approach, focusing on the description of the observed trends without speculative interpretations. Additionally, we have carefully incorporated all your suggested changes throughout the manuscript, figures, and tables. To ensure transparency and facilitate a comprehensive review, we have highlighted the modifications made in the revised version of the manuscript. By following your guidance, we believe our manuscript now offers a more accurate representation of our findings while ensuring a rigorous and unbiased scientific approach. We have also included a section in the revised manuscript that highlights the limitations of our study and the need for further research to investigate potential causative factors more comprehensively. We hope that you find the revised version more suitable and aligned with the rigorous standards of the journal.

Q: To the best of my understanding, the authors’ objective was to describe cohorts of COVID-19 patients using age, sex, disease outcome and hematological parameters during multiple waves of the pandemic. If this is the case, the current statistical analysis is appropriate. However, the authors should reconsider their phrasing to avoid making associations between specific waves and changes in hematological parameters throughout the manuscript.

A: We would like to express our gratitude for your thoughtful comment regarding the aim of our study. Your understanding aligns perfectly with our intentions for this research. Indeed, our primary objective was to describe cohorts of COVID-19 patients, analyzing age, gender, disease outcome, and hematological parameters across multiple waves of the pandemic. We acknowledge the importance of clarity in scientific writing, and your comment has prompted us to critically reconsider our phrasing throughout the manuscript. We understand that it is essential to avoid making unwarranted associations between specific waves and changes in hematological parameters, as this might inadvertently introduce bias or misinterpretation. In light of your valuable feedback, we have carefully revised the manuscript to ensure that our language emphasizes a descriptive approach rather than attempting to establish causative relationships or identify significant associations between specific waves and hematological changes. Our focus now remains on presenting the data and observations in a neutral and unbiased manner. We sincerely apologize for any ambiguity in our previous phrasing, and we thank you for bringing this to our attention. The revised version of the manuscript has been updated to reflect these improvements. Once again, we appreciate your diligence in reviewing our work and providing valuable suggestions that have undoubtedly strengthened the clarity and scientific rigor of our study.

Q: If the authors would like to make associations between their observations and specific waves of COVID-19, they should consider undertaking a matched analysis based on age, sex, disease outcome, and the severity of symptoms at time of hospital admission (if available). By matching patients with similar characteristics, the authors could control for potential confounding factors and improve the robustness of their analysis.

A: We would like to express our gratitude for your insightful comment regarding the potential associations between our observations and specific waves of COVID-19. Your suggestion to undertake a matched analysis based on age, sex, disease outcome, and symptom severity at the time of hospital admission is undoubtedly valuable for improving the robustness of our analysis. We have carefully considered your recommendation and understand the benefits of controlling for potential confounding factors. However, we wish to clarify our primary objective for this study. Our aim is not to establish causal relationships or make direct associations between specific waves of COVID-19 and changes in hematological parameters. Due to the limitations in our data and the retrospective nature of our study, conducting a matched analysis to control for potential confounders is challenging. Instead, we are primarily interested in exploring the epidemiological characteristics of COVID-19 patients, specifically focusing on age, sex, disease outcome, and hematological parameters, across multiple waves of the pandemic. By describing the cohorts of patients in each wave, we aim to identify trends and potential patterns in the data. We have thoroughly revised the manuscript to emphasize our intent and avoid any misinterpretation. We have carefully clarified throughout the text that our analysis does not permit making direct causal associations, and we now emphasize the descriptive nature of our study. We greatly appreciate your guidance in enhancing the clarity and scope of our research. Your thoughtful feedback has been invaluable in refining the manuscript to align more closely with our study's objectives and limitations.

---

## [Decision Letter · Decision Letter 3]

7 Aug 2023

Evaluation of Hematological Parameters Alterations in Different Waves of COVID-19 Pandemic: A Cross-Sectional Study

PONE-D-22-21644R3

Dear Dr. Nakhaie,

We’re pleased to inform you that your manuscript has been judged scientifically suitable for publication and will be formally accepted for publication once it meets all outstanding technical requirements.

Kind regards,

Bhaswati Chatterjee, PhD

Academic Editor

PLOS ONE

Additional Editor Comments (optional):

Reviewers' comments:

Reviewer's Responses to Questions

**Comments to the Author**

1. If the authors have adequately addressed your comments raised in a previous round of review and you feel that this manuscript is now acceptable for publication, you may indicate that here to bypass the “Comments to the Author” section, enter your conflict of interest statement in the “Confidential to Editor” section, and submit your "Accept" recommendation.

Reviewer #2: All comments have been addressed

Reviewer #3: All comments have been addressed

2. Is the manuscript technically sound, and do the data support the conclusions?

Reviewer #2: Yes

Reviewer #3: Yes

3. Has the statistical analysis been performed appropriately and rigorously? 

Reviewer #2: Yes

Reviewer #3: Yes

4. Have the authors made all data underlying the findings in their manuscript fully available?

Reviewer #2: Yes

Reviewer #3: Yes

5. Is the manuscript presented in an intelligible fashion and written in standard English?

Reviewer #2: Yes

Reviewer #3: Yes

6. Review Comments to the Author

Reviewer #2: (No Response)

Reviewer #3: (No Response)

7. PLOS authors have the option to publish the peer review history of their article (what does this mean?). If published, this will include your full peer review and any attached files.

Reviewer #2: No

Reviewer #3: **Yes: **DEMANOU Maurice

---

## [Editor Report · Acceptance letter]

16 Aug 2023

PONE-D-22-21644R3 

Evaluation of Hematological Parameters Alterations in Different Waves of COVID-19 Pandemic: A Cross-Sectional Study 

Dear Dr. Nakhaie:

I'm pleased to inform you that your manuscript has been deemed suitable for publication in PLOS ONE. Congratulations! Your manuscript is now with our production department. 

Kind regards, 

on behalf of

Dr. Bhaswati Chatterjee 

Academic Editor

PLOS ONE